# The Interference of RNA Preservative and Post-Collection Interval on RNA Integrity from Different Mice Tissues

**DOI:** 10.3390/genes16121421

**Published:** 2025-11-28

**Authors:** Ting Xie, Hui Zhu, Xiaoxi Wang, Fangyuan Li, Anqi Wang, Yaran Zhang, Sumei Zhang, Dan Guo

**Affiliations:** Clinical Biobank, National Infrastructures for Translational Medicine, Peking Union Medical College Hospital, Chinese Academy of Medical Sciences, Beijing 100730, China; xietingpumch@163.com (T.X.); zhuhuiwit@163.com (H.Z.); wangxx_0307@163.com (X.W.); doris_lfy@163.com (F.L.); anqi620@163.com (A.W.); yaranyaran@163.com (Y.Z.); zsm_1218@163.com (S.Z.)

**Keywords:** RNA preservative, RNAlater, post-collection interval, RNA integrity number (RIN), Biobank

## Abstract

Background: For precise and reliable gene expression analysis, the acquisition of high-quality RNA is contingent upon excellent tissue preparation and handling. The optimal method for preserving tissues after surgical resection remains challenging due to the delays in delivery or the absence of cold storage equipment. Although RNAlater has been extensively adopted for tissue preservation, few studies have systematically evaluated the effects of various tissue preservation solutions and post-collection intervals on RNA integrity across a range of tissue types. Methods: Ten types of mouse tissues, representing common tissue species in biobanks, were collected after resection. Tissues were either flash-frozen in liquid nitrogen as controls or immersed in one of three RNA preservatives—TRIzol and two commercial RNAlater solutions—and stored at room temperature (RT) for 0, 4, or 8 h before being frozen. Total RNA was extracted using TRIzol method, and its integrity was assessed using the RNA Integrity Number (RIN). Results: The results indicated that both the post-collection interval and the type of RNA preservative significantly impact RNA integrity. Pancreatic tissue showed the poorest RNA integrity (RIN < 5.5), whereas heart and ovary tissue yielded high-quality RNA (RIN > 7) even without any preservatives after 8 h at RT. To maintain baseline RNA integrity (RIN > 5.5), tissues including brain, kidney, muscle, liver, intestine, and uterus should be immersed in preservative and frozen within 8 h. For lung tissue preserved in RNAlater, the maximum recommended time at RT was 4 h. Conclusions: Robust, high-quality RNA can be obtained from most mouse tissues stored in RNA preservatives for up to 8 h at RT, with only minor variations observed across the different preservatives tested.

## 1. Introduction

High-quality tissues with preserved biomolecule integrity ensure the precision and reproducibility of downstream research applications. In particular, the extraction of high-quality total RNA from tissue is critical for advanced transcriptomic analyses, such as RNA hybridization, quantitative real-time fluorescent polymerase chain reaction (qRT-PCR), bulk RNA microarrays, RNA-seq, and spatial transcriptomics [1]. The Agilent Bioanalyzer RNA integrity number (RIN) is widely regarded as the gold standard for assessing the quality of RNA isolated from all biological sources [2]. Nonetheless, the integrity of RNA isolated from tissue samples is susceptible to a range of pre-analytical variables. These include ischemia duration [3,4,5] and the type of preservation solution [6,7,8,9], as well as repeated freeze–thaw cycles and extended storage period [10,11,12]. Although previous studies have investigated the effects of RNA preservative types [8,13], ischemia time [4], and tissue type [14] on RNA integrity, their simultaneous examination has been relatively scarce. In a study we previously undertook, the effects of ex vivo ischemia time on RNA integrity and gene expression were evaluated in 18 carcinoma tissues spanning 6 cancer types. We found that processing fresh tissues within 2 h is critical to avoid RNA degradation [15].

Biobanks are dedicated to the collection, pretreatment, storage and distribution of biospecimens, thereby facilitating basic, translational and clinical research. They play an important role throughout the entire spectrum of disease prevention, treatment, follow-up, and therapy development [16]. For most biobanks, flash-freezing tissues in liquid nitrogen is regarded as the gold standard for preservation. Human tissues are usually sourced from biopsies, surgical resections, and donated cadavers [17]. In clinical practice, the immediate cryopreservation of tissue samples in liquid nitrogen is often unfeasible owing to complex surgical procedures, a limited supply of liquid nitrogen, and logistical challenges in timely transportation. Therefore, as a complement to flash-freezing in liquid nitrogen, immersion in chemical stabilizers or preservatives represents a widely used approach for tissue preservation. The conventional TRIzol reagent is frequently selected for this purpose, owing to its established efficacy and reliability in total RNA extraction [18]. Two additional RNAlater reagents from Invitrogen and Sigma, which are commonly used in biomedical studies, were also selected for this investigation. RNAlater is an aqueous, non-toxic reagent designed to rapidly permeabilize tissues and stabilize intracellular RNA by inactivating RNase upon collection. This action preserves RNA integrity without requiring immediate processing or frozen storage [19]. Although the manufacturer advises overnight storage at 2–8 °C before long-term freezing, the impact of this specific step compared to immediate freezing will be examined in this research.

We employed mouse tissues in this study due to the dual advantages of this model: the ability to maintain controlled conditions and genetic uniformity, which circumvent the challenges of obtaining high-quality fresh human tissues. We selected the most commonly stored tissue types in biobank including liver, pancreas, intestine, heart, lung, uterus, ovary, muscle, brain, and kidney to investigate the influence of post-collection intervals and types of RNA preservatives on tissue RNA integrity. This article aims to provide experimental evidence to help clinical researchers optimize tissue quality during sample collection. Specifically, it offers guidance on selecting appropriate preservatives and optimizing delivery procedures based on tissue characteristics. To this end, our study seeks to (1) compare the effects of different RNA preservatives on RNA integrity; (2) quantify the impact of short-term storage at room temperature (RT) on RNA integrity; and (3) develop tailored preservation protocols by identifying the optimal combination of preservative and post-collection interval for diverse tissue types.

## 2. Materials and Methods

### 2.1. Collection of Mice Tissues

8-week-old C57BL/6J female mice were used in these experiments. Tissue samples for RNA extraction were harvested from five mice, with each mouse contributing ten distinct tissue types to serve as biological replicates. Mice were euthanized with Tribromoethanol, and then tissues (including liver, pancreas, intestine, heart, lung, uterus, ovary, muscle obtained from the hind leg, brain, and kidney) were collected. Each tissue sample was dissected into approximately 0.4 × 0.4 × 0.4 cm fragments (~40 mg). These fragments were immediately immersed in one of the following RNA preservative conditions: none (fresh control), TRIzol (Invitrogen, San Diego, CA, USA), RNAlater (Invitrogen, Lot 2817037) and RNAlater (Sigma, St. Louis, MI, USA, Lot MKCR6534). Tissues were maintained at RT in these solutions for 0 h (control), 4 h, or 8 h, after which all samples were flash-frozen in liquid nitrogen. Finally, RNA was extracted simultaneously from all tissues across all time points and preservative conditions using a uniform protocol.

### 2.2. RNA Extraction and Evaluation of RNA Integrity

Total RNA was isolated with TRIzol reagent (Invitrogen, San Diego, CA, USA) following the manufacturer’s instructions. Tissues previously immersed in Invitrogen or Sigma RNA preservatives were first washed three times with PBS to remove the residual solution. Subsequently, each sample was homogenized with 1 mL of TRIzol^®^ reagent (Invitrogen; Thermo Fisher Scientific, Waltham, MA, USA) with a homogenizer (Tiangen, Beijing, China, TGuide H24R). The RNA extraction was then performed by adding 200 μL of chloroform, followed by centrifugation. The aqueous phase was collected, and 500 μL of isopropyl alcohol was added to precipitate the RNA. The resulting RNA pellets were washed twice with 75% ethanol, air-dried at RT, and finally dissolved in RNase-free water before stored at −80 °C. RNA integrity was analyzed using the Agilent 4200 tape station using Agilent RNA 6000 nano kit (Agilent Technologies, Santa Clara, CA, USA). The RIN value, which ranges from 1 (highly degraded) to 10 (fully intact), was automatically calculated by the accompanying Agilent software.

### 2.3. Statistical Analysis

All statistical analyses were performed using GraphPad Prism 6.0 (GraphPad Software, La Jolla, CA, USA). The data are represented as means ± SEM. We assessed the statistical relationships between RIN values and post-collection intervals using Spearman’s rank correlation coefficient for Figure 1. Multiple-group comparisons were performed using one-way analysis of variance (ANOVA) followed by Tukey’s post hoc test. For comparisons involving more than two variables, two-way ANOVA followed by Bonferroni’s post hoc test was used. A *p*-value of less than 0.05 was considered statistically significant for all analyses.

## 3. Results

### 3.1. Correlation of RNA Integrity with Post-Collection Intervals

To independently assess the impact of post-collection intervals on RIN value, tissue samples were maintained at RT without any preservatives for 0 h, 4 h and 8 h. Significant negative correlations between RNA integrity and post-collection intervals were observed in muscle, intestine, pancreas, kidney and liver (Figure 1F–J). Although a similar declining trend was noted in the heart, brain, ovary, uterus, and lung (Figure 1A–E), the correlation in these tissues did not reach statistical significance.

### 3.2. Effective Preservation: Overnight Immersion or Immediate Freezing?

According to the protocols from different RNAlater brands, samples are typically incubated at 4 °C overnight prior to freezing to facilitate thorough penetration of the preservative into the tissue. However, it remains unclear whether this step is necessary when liquid nitrogen is readily available. To address this, we systematically compared the effect of overnight storage at 4 °C versus immediate freezing in liquid nitrogen on RNA integrity. As a result, no significant difference was observed between the two treatments across both RNAlater brands (Figure 2). Moreover, although not statistically significant, most tissues preserved in RNAlater exhibited a trend toward higher RNA integrity when flash-frozen immediately rather than stored overnight at 4 °C. These findings indicate that the overnight incubation step is unnecessary and that samples submerged in RNAlater can be directly flash-frozen in liquid nitrogen.

### 3.3. Evaluation of Preservative Efficacy in Maintaining RNA Integrity in Various Tissues

To further explore the protective effects of different RNA preservatives, we submerged ten mouse tissues in various commercial solutions for 0, 4, and 8 h, followed by RNA extraction using the TRIzol method and RIN measurement. As shown in Figure 3A, the RIN values of the heart were the highest among all the tissues examined. Ovarian tissue also displayed high integrity (Figure 3C), indicating that heart and ovarian tissue may be stored at RT for up to 8 h without preservative if liquid nitrogen is unavailable. Conversely, pancreatic tissue showed the lowest RIN values (Figure 3B). For other seven tissues (Figure 3D–G), immediate submersion in RNA preservatives could protect the RNA integrity to different extend at RT. Muscle and kidney, which started with high RIN values (>7) without preservative, then dercrased from the 0 h baseline, but maintained this lever sustained RNA integrity to at least the 7 baseline significantly over 8 h with adding preservative. In tissues with intermediate initial RIN (5.5–7), such as brain, liver, and uterus, RNA degradation was attenuated by preservatives within the first 4 h. Even in intestines and lungs, which had low starting RIN (<5.5), adding a preservative raised and sustained RNA integrity to at least the 5.5 baseline over 4 h. Furthermore, brand-specific performance trends were observed: Invitrogen RNAlater appeared more effective for lung tissue, whereas Sigma RNAlater seemed better suited for intestine and uterus (Figure 3H–J). However, as these trends were not statistically significant, no firm conclusions can be drawn without follow-up studies with larger sample sizes.

## 4. Discussion

Although inherent biological variability within bio-specimens influences gene expression results [20], the reliability of gene expression profiling for downstream studies depends primarily on RNA quality and how well tissue or cell samples are preserved during the biobank process [21]. Using compromised RNA can severely undermine the outcomes of subsequent experiments—such as qRT-PCR, RNA-seq, and spatial transcriptomics—which are often costly and time-consuming. Studies have shown that RNA integrity primarily affects the accuracy and reproducibility of qRT-PCR results, rather than the amplification efficiency of the PCR reaction. It is recommended that an RIN above 5 represents good RNA quality, while an RIN above 8 is considered ideal for downstream analyses [22]. In RNA-seq studies, gene expression measurements are frequently influenced by RIN, which also exhibits a negative correlation with post-sequencing quality metrics [23]. An RIN threshold of 7 is widely adopted in RNA-seq as a benchmark for adequate RNA quality [24,25]. For spatial transcriptomics, one study reported no significant differences in sequencing results between samples with 6 ≤ RIN < 7 and those with RIN ≥ 7, suggesting that an RIN cutoff above 6 may be acceptable for quality control [26]. Recently, the spatial RNA integrity number (sRIN) assay has been developed to evaluate RNA integrity at each spatial location, overcoming the limitation of bulk RIN measurements that only reflect an average across heterogeneous regions. In this approach, areas with sRIN ≥ 7.5 are deemed suitable for subsequent high-cost spatial analyses [27]. Accordingly, in this study, RIN cutoffs of 5.5 and 7 were applied, reflecting the requirements of most downstream applications.

It is postulated that the relative stability of RNA in cardiac tissues stems from inherently low levels of endogenous ribonucleases, coupled with structural features that provide a shield from exogenous ribonucleases derived from bacteria or other sources [28]. Another study demonstrated that heart RNA exhibited significantly greater stability than that from the brain, despite both organs being well shielded from the external environment [14]. The relatively high RNA integrity observed in ovarian tissue may be related to its critical reproductive role. In a study focusing on testicular tissue—also relevant to fertility—researchers found that indicators of RNA quality such as the 28S/18S ratio remained favorable even after one year of storage in RNAlater at 4 °C [29]. In contrast, due to the inherent characteristics of the pancreas, the protective effect of RNA preservatives is very limited. Extracting intact RNA from pancreatic tissue is challenging owing to the abundance of endogenous ribonucleases; during dissection, pancreatic acinar cell granules are disrupted, releasing their contents and triggering tissue autolysis, which rapidly degrades RNA. However, perfusing the pancreas with RNAlater via the bile duct prior to tissue dissection and subsequent rinsing in the preservative can help obtain high-quality RNA with RIN values above seven [30]. Consistent with established knowledge, RNA degrades more rapidly in tissues with high ribonuclease levels—such as the liver, pancreas, spleen, kidneys, and blood cells—compared to tissues with low ribonuclease content, including the heart, brain, and teeth [28]. Correspondingly, our data illustrated that tissues with lower ribonuclease activity (e.g., brain, heart, ovary, lung, and uterus) generally showed better RNA stability than those rich in ribonucleases (e.g., pancreas, intestine, kidney, and muscle). For most tissue types, the use of TRIzol or RNAlater can markedly enhance RNA integrity during short-term storage.

RNAlater is designed to rapidly permeate tissues and immediately inactivate RNases, thereby stabilizing cellular RNA and preserving RNA integrity upon collection. The manufacturer recommends storing samples at 2–8 °C overnight before freezing. For both types of RNAlater, however, no significant difference in RNA quality was observed between samples stored overnight at 4 °C and those flash-frozen directly in liquid nitrogen. A possible explanation is that the relatively small size of mouse tissue blocks in this study allows for rapid and thorough penetration of the reagent, making overnight immersion unnecessary. For larger tissue specimens, however, this step may still be indispensable, or alternatively, tissues may be sectioned into smaller fragments prior to cryopreservation. Therefore, if liquid nitrogen or dry ice is available, it is preferable to immediately freeze tissue samples—provided they are not too large—after immersion in RNAlater, rather than storing them overnight at 2–8 °C.

The commonly used TRIzol method yields higher RNA quantities compared to column-based purification when processing small tissue pieces [31]. Additionally, increasing mRNA concentration in solution has been shown to enhance its stability [7]. Therefore, if a tissue sample is specifically intended for RNA-related studies, TRIzol is a suitable option for on-site preservation, as it can be directly used for RNA extraction and eliminates the need for washing steps to remove preservatives like RNAlater. It should be noted, however, that completely removing RNAlater requires multiple thorough PBS washes, and any residual solution may affect RNA yield and purity. In clinical practice, the specific downstream applications of a tissue sample are often not predetermined at the time of collection. In such cases, a multipurpose preservative such as RNAlater offers a distinct advantage, as it stabilizes RNA while also maintaining tissue integrity for histological analysis [32,33] and proteomic study [34,35,36]. One study showed that RNAlater preservation for 7–14 days was suitable for histological characterization of mouse lung tissue, though not for brain, liver, or kidney under the same conditions [33]. However, it is important to note that previous research has also indicated that RNAlater may interfere with gene expression studies; for example, genes with high GC content showed higher expression levels in flash-frozen samples compared to those stored in RNAlater [19].

In our previous research with 18 carcinoma tissues, we found that fresh tissues should be processed within 2 h to prevent RNA degradation and structural alterations [15]. Similarly, a study on protective strategies for frozen mouse brain sections in single-cell and spatial transcriptomics reported that samples left at RT for 0–8 h experienced a decline in median RIN value from 8.1 to 2.3. After just 2 h, the median RIN had already dropped below 4.4. Furthermore, while storage at RT for up to 4 h did not affect the stability of common reference genes, extending this period to 8 h significantly reduced the expression abundance of *Gapdh*, *Actb*, and *Hprt1* [37]. Based on these findings, we recommend that tissue samples not treated with RNA preservatives be processed and frozen within 2 h post-resection. This study provides practical guidance for scenarios involving delayed sample processing due to surgical complexity, limited access to liquid nitrogen, or other logistical challenges.

Although mouse models have inherent limitations in fully elucidating human disease mechanisms, they offer significant advantages for biomedical research, including controlled experimental conditions and genetic uniformity. These features help overcome the practical challenges associated with obtaining high-quality fresh human tissues. A study comparing transcriptome-wide mRNA decay datasets between humans and mice revealed a strong conservation of mRNA half-lives, with a Pearson correlation coefficient of 0.78. This indicates that mRNA turnover rates are more evolutionarily conserved than previously recognized, despite approximately 75 million years of divergence between these mammalian species [38]. While findings from mouse tissues provide valuable insights, these findings require further validation and extension in human samples. We recommended that biobanks that have implemented a quality management system compliant with the ISO 20387 standard systematically analyze quality control results from biospecimens and their performance in downstream research applications. Based on the requirements of specific clinical research questions and intended analytical workflows, standardized yet adaptable SOPs should be established for clinical sample collection, processing, storage, and subsequent analysis. Such practices will help build a robust quality assurance framework, enhance research reproducibility, and ultimately contribute to the development of industry-wide guidelines and consensus.

## 5. Conclusions

Consistent with previous reports, delayed preservation compromised RNA integrity in most tissue types examined. Notably, heart and ovarian tissues—which exhibit inherently low RNase activity—were relatively stable without any preservative at RT. Our results demonstrate that robust, high-quality RNA can be obtained from most mouse tissues stored in RNA preservatives for up to 8 h, with only minor variations observed among the different preservatives tested. This study provides guidance for establishing detailed, tissue-specific standard operating procedures in clinical biobanks and supports the stratified management and quality assurance of biospecimens.

## Figures and Tables

**Figure 1 genes-16-01421-f001:**
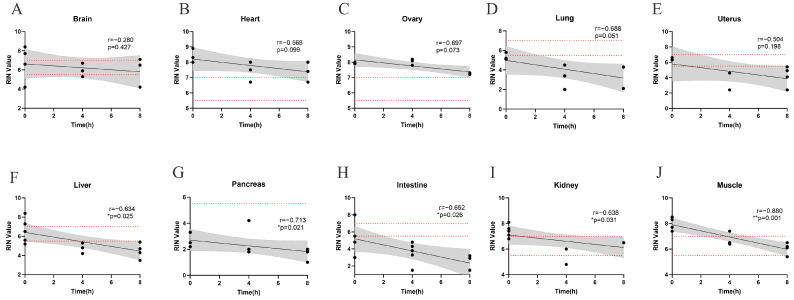
Linear regression lines for each tissue between RIN value and post-collection intervals with 95% confidence limits. The distribution of samples with significant correlations ((**F**–**J**) liver, pancreas, intestine, kidney and muscle) compared to those with no correlation ((**A**–**E**) brain, heart, ovary, lung, and uterus) is shown. The dash lines indicated the RIN cutoffs of 5.5 and 7, reflecting the requirements of most downstream applications. The statistical relationships between RIN value and post-collection intervals were assessed using Spearman’s rank correlation coefficient, with *p* < 0.05 considered statistically significant. (*, *p* < 0.05; **, *p* < 0.01).

**Figure 2 genes-16-01421-f002:**
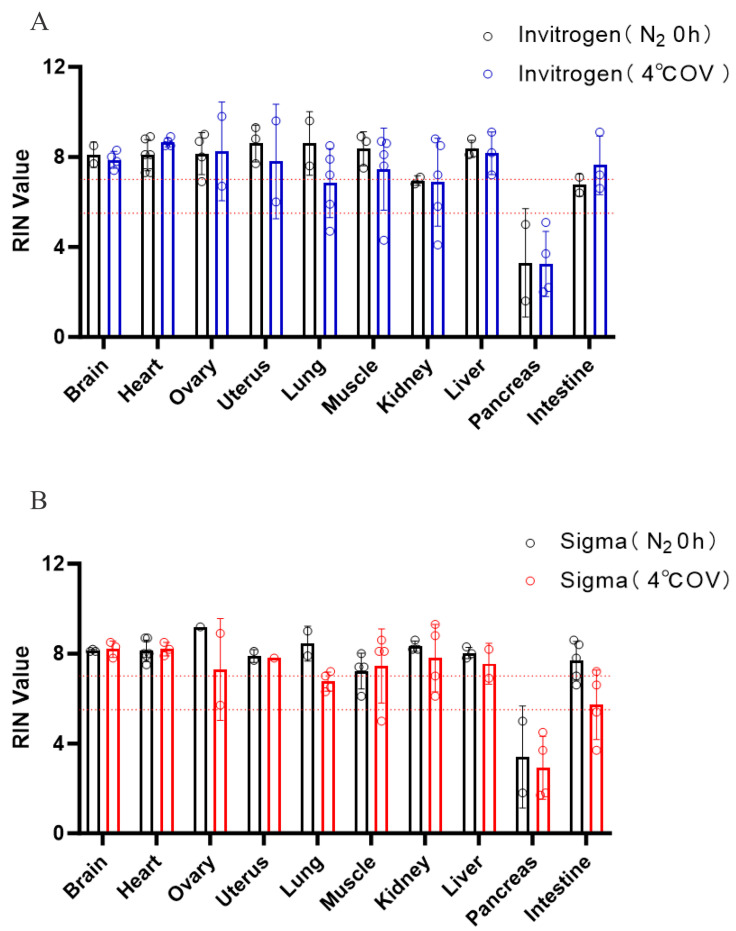
The effect of overnight storage on the integrity of RNA. RIN values from tissues immersed in RNAlater ((**A**) Invitrogen) or RNAlater ((**B**) Sigma) with flash freezing in liquid nitrogen or keeping overnight at 4 °C were assessed. The dash lines indicated the RIN cutoffs of 5.5 and 7, reflecting the requirements of most downstream applications. Data represent means ± SEM.

**Figure 3 genes-16-01421-f003:**
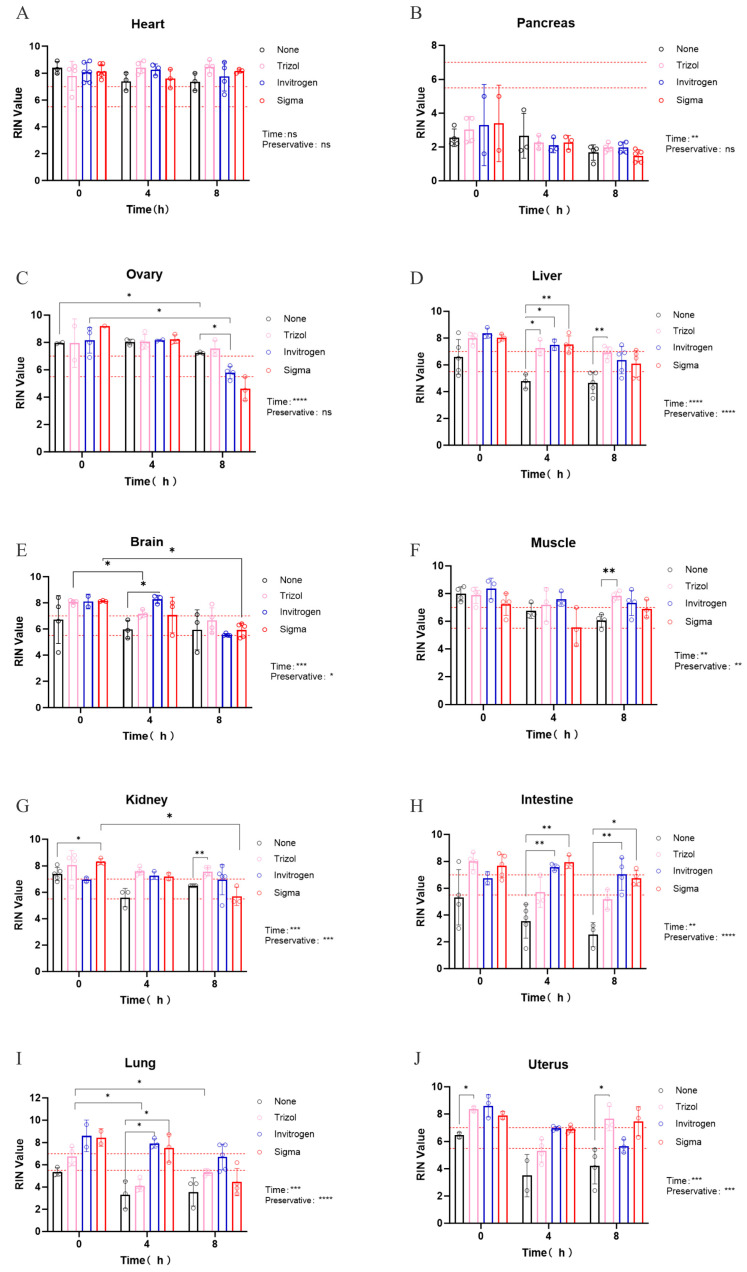
The protection effect of different RNA preservatives (None, TRIzol, Invitrogen RNAlater, Sigma RNAlater) on the RIN values from ten mouse tissues. (**A**) Heart (*p*-values for time and preservative: 0.700, 0.395), (**B**) Pancreas (*p*-values for time and preservative: 0.006, 0.975), (**C**) Ovary (*p*-values for time and preservative: <0.0001, 0.470), (**D**) liver (*p*-values for time and preservative: <0.0001, <0.0001), (**E**) Brain (*p*-values for time and preservative: 0.003, 0.042), (**F**) Muscle (*p*-values for time and preservative: 0.0028, 0.0029), (**G**) Kidney (*p*-values for time and preservative: 0.001, 0.0007), (**H**) Intestine (*p*-values for time and preservative: 0.0034, <0.0001), (**I**) Lung (*p*-values for time and preservative: 0.0034, <0.0001), (**J**) Uterus (*p*-values for time and preservative: 0.0003, 0.0002). The dash lines indicated the RIN cutoffs of 5.5 and 7, reflecting the requirements of most downstream applications. Data are presented as means ± SEM (*, *p* < 0.05; **, *p* < 0.01; ***, *p* < 0.001, ****, *p* < 0.0001).

## Data Availability

The original contributions presented in the study are included in the article, further inquiries can be directed to the corresponding author.

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
