# Peer review of "The Interference of RNA Preservative and Post-Collection Interval on RNA Integrity from Different Mice Tissues"

_genes, 2025, doi:10.3390/genes16121421_

Round 1

Reviewer 1 Report

Comments and Suggestions for Authors

Regarding #genes-3980191 "The Interference of RNA Preservative and Post-Collection Interval on RNA Integrity from Different Mice Tissues"

This is a manuscript that explores the impact of RNA collection methods and post-collection intervals on RNA integrity in different mouse tissues. The study aims to serve as a preanalytical tissue handling guideline for clinical researchers within the field of RNA expression-based research. For this, the authors used five mice as biological replicates, on which they tested two different RNA collection methods (Trizol and RNAlater) at two different timepoints (4 & 8h) besides baseline (0h). The authors found that prolonged degradation time of tissues impacted RNA integrity in a tissue specific manner.

The authors suggest that their study serves ‘to provide experiments evidence for clinical researchers’ (line 74), although RNA degradation rates are not uniform across all species. The authors should modify their ‘hypothesis’ according to that.

It is not clear to me, why the authors compare Sigma and Invitrogen RNAlater’s effect on RNA integrity, as there are no fundamental differences in the product's composition or function between the two brands. Given that, the authors should include a scientific based argument for why they designed an experimental study comparing two brands of the sampe product. Hence, it is not surprising that the authors find no significant difference in RNA integrity between the two, as shown in Fig.3.

Even though, the authors find no statistical difference, they still argue that one brand is preferred over the other for collection of some tissues:

Line154-158: Furthermore, as shown in the figure 2 I-J, Invitrogen RNAlater is more proper for the temporary storage of lung tissue and Sigma RNAlater is suit for intestine and uterus tissue at three time points compared to other RNA preservatives, although no statistically significant difference was observed.

Comment: The authors should correct their statement so that it is not implied that one brand is preferred over the other. Alternatively, the authors could argue, that they ‘observe a trend for tissue J & I’, e.g., to be explored in a follow-up study using a larger sample size.

In line 50 the authors argue that: ‘It is suggested that fresh tissues should be processed within 2 h to avoid RNA degradation. [14].’ However, the authors do not include the 2h time-point in their experimental set-up. The authors should address their reasoning for leaving out this time-point in their experimental set-up that besides baseline (0h) only has 4h and 8h, respectively, as additional timepoints.

Line 57-58: ‘Given that most clinical tissues cannot be immediately pretreated and storage in liquid nitrogen in biobank lab upon collection due to the surgery complexity, shortage of liquid nitrogen and lack of timely delivery.’

The authors should more clearly argue for the clinical relevance of choosing 4h and 8h as time points in their experimental set-up. It is, e.g., difficult for me to imagine that tissue from surgical procedures is left for 8 hours upon collection, unless we are talking about post-mortem samples, which is not implied by the authors..

Line 38-40 In particularly, high-quality total RNA extracted from tissue is pivotal for advanced transcriptomic profiles using RNA molecular, such as RNA hybridization, real-time fluorescent quantitative polymerase chain reaction (qRT-PCR), bulk RNA microarrays, RNA-seq, and recently emerged spatial transcriptomics [1].

Comment: Regarding ‘recently emerged spatial transcriptomics’: The concept of spatial transcriptomics is old (+50 years) while combining in situ hybridization with Next-generation-Sequencing for a high-resolution view of the transcriptome tracks nine years back. The latter relies on e.g., machine learning to "deconvolute" bulk RNA-seq data for prediction of spatial gene expression. Maybe the authors could rephrase to ensure clarity for the readers?

Sometimes sentences seem out of context and appear to be without references, e.g.,:

  • Line 46-50: Only a few studies have compared the impact of both post-collection interval and different RNA preservative [8] on the RNA integrity in different tissues [13]. In a previous study, we surveyed the effect of ex vivo ischemia time on RNA integrity and expression of genes isolated from 18 carcinoma tissues of 6 types of cancer. It is suggested that fresh tissues should be processed within 2 h to avoid RNA degradation. [14].
  • Line 171-172: Researchers found that RNA integrity affected the qRT-PCR performance, not the PCR efficiency. It recommends a RIN higher than 5 as good quality and higher than 8 as perfect for downstream application.

Comment: the authors need to rephrase the manuscript to ensure clarity and coherence for the reader and ensure that proper references are applied following scientific arguments stated in sentences by the authors.

The manuscript contains a high number of typos, and the grammar is very poor, e.g.:

  • Line 57-58: Given that most clinical tissues cannot be immediately pretreated and storage in liquid nitrogen in biobank lab upon collection due to the surgery complexity, shortage of liquid nitrogen and lack of timely delivery.
  • Line 63-64: Other two RNAlaters from Invitrogen and Sigma frequently adopted in biomedical studies were selected in this study.
  • Line 213: On the other hand, if the tissue sample has not been decided to conduct what kind of research at the collection in most clinical occasions.
  • Line 254-255: Consistent with previous reports, delayed preservation time has a negative correlation with RNA integrity for most tissue species.

Comment: it needs professional proof reading.

In the discussion, the authors elaborate on the introduction with poor relevance to their own study: in line 179-191 the authors e.g. elaborate on spatial transcriptomics – an analysis the authors did not perform in their study. The authors could benefit from elaborating on the discussion of their own results. In general, the discussion is difficult to follow, and seems to lead to the promotion of the authors ‘upcoming quality management system’.

Comment: should this paper be accepted, the discussion needs to be revised with the purpose of promoting consistency between the relevance of the findings of their study on other studies discussed.

The authors do not directly refer to their results in the discussion. To improve the reading experience the authors should refer to relevant figure numbers in the discussion following sentences referring to their results.

Figure 3 should be revised, so that the colors indicating ‘Trizol’ and ‘Sigma’ appear more distinctive.

Author Response

Point-by-point response to previous review comments

Dear Editors and Reviewers:

Thank you for your letter and for the Reviewers’ comments concerning our manuscript entitled “The Interference of RNA Preservative and Post-Collection Interval on RNA Integrity from Different Mice Tissues” (genes-3980191). Those comments are all valuable and very helpful for revising and improving our paper, as well as being of great importance and guiding significance to our research. We have carefully studied the comments and made corrections that we hope will meet with approval. The revised portion is marked in the revised manuscript. The spelling and grammar of the article have been professionally proofread and polished, with all revisions clearly highlighted in yellow. The main corrections and the point-by-point response to the reviewers’ comments are as follows:

Reviewer 1

This is a manuscript that explores the impact of RNA collection methods and post-collection intervals on RNA integrity in different mouse tissues. The study aims to serve as a preanalytical tissue handling guideline for clinical researchers within the field of RNA expression-based research. For this, the authors used five mice as biological replicates, on which they tested two different RNA collection methods (Trizol and RNAlater) at two different timepoints (4 & 8h) besides baseline (0h). The authors found that prolonged degradation time of tissues impacted RNA integrity in a tissue specific manner.

The authors suggest that their study serves ‘to provide experiments evidence for clinical researchers’ (line 74), although RNA degradation rates are not uniform across all species. The authors should modify their ‘hypothesis’ according to that.

Response 1:Thank you for your insightful question. Indeed, RNA degradation rates vary across species. However, a study compiling 39 humans and 27 mouse transcriptome-wide mRNA decay datasets revealed strong conservation. Among one-to-one orthologs, the consensus mRNA half-lives between humans and mice showed a high Pearson correlation of 0.78. This indicates that mRNA half-life is more strongly conserved than previously thought, despite the ~75 million years of evolution separating these mammalian species (Agarwal and Kelley 2022). (Line271-276)

It is not clear to me, why the authors compare Sigma and Invitrogen RNAlater’s effect on RNA integrity, as there are no fundamental differences in the product's composition or function between the two brands. Given that, the authors should include a scientific based argument for why they designed an experimental study comparing two brands of the sampe product. Hence, it is not surprising that the authors find no significant difference in RNA integrity between the two, as shown in Fig.3.

Even though, the authors find no statistical difference, they still argue that one brand is preferred over the other for collection of some tissues:

Line154-158: Furthermore, as shown in the figure 2 I-J, Invitrogen RNAlater is more proper for the temporary storage of lung tissue and Sigma RNAlater is suit for intestine and uterus tissue at three time points compared to other RNA preservatives, although no statistically significant difference was observed.

Comment: The authors should correct their statement so that it is not implied that one brand is preferred over the other. Alternatively, the authors could argue, that they ‘observe a trend for tissue J & I’, e.g., to be explored in a follow-up study using a larger sample size.

Response 2:We thank the reviewer for raising this important point. We fully acknowledge that our initial wording could be misinterpreted as suggesting a brand preference, which was not statistically supported. The selection of these two brands was made mainly for the purpose of comparing the RNAlaters that are most widely available and used in clinical biobanking, thereby providing an objective reference for users. We have carefully modified the manuscript to present this as a practical comparison without any claims of superiority, ensuring our interpretation is accurate and balanced. (Line170-171)

In line 50 the authors argue that: ‘It is suggested that fresh tissues should be processed within 2 h to avoid RNA degradation. [14].’ However, the authors do not include the 2h time-point in their experimental set-up. The authors should address their reasoning for leaving out this time-point in their experimental set-up that besides baseline (0h) only has 4h and 8h, respectively, as additional timepoints.

Response 3:We sincerely thank the reviewer for this valuable feedback. The 2-hour time point was not included in our experimental design. This decision was based on our previous finding [14] that fresh tissues should be processed within 2 hours to prevent significant RNA degradation, implying that RNA integrity remains largely stable and acceptable within this window for most tissues. Consequently, the present study was specifically designed to evaluate the impact of preservation methods and longer post-collection intervals (4h and 8h) on RNA integrity, focusing on scenarios where immediate processing is not feasible.

Line 57-58: ‘Given that most clinical tissues cannot be immediately pretreated and storage in liquid nitrogen in biobank lab upon collection due to the surgery complexity, shortage of liquid nitrogen and lack of timely delivery.’

The authors should more clearly argue for the clinical relevance of choosing 4h and 8h as time points in their experimental set-up. It is, e.g., difficult for me to imagine that tissue from surgical procedures is left for 8 hours upon collection, unless we are talking about post-mortem samples, which is not implied by the authors.

Response 4:We appreciate the opportunity to clarify our choice of time points. As the comment rightly points out, while the standard practice is to process surgical tissues within 2 hours (~80% tissues in our biobank), logistical constraints within an 8-hour workday—such as complex surgeries or staffing shortages—can inevitably delay processing for some samples. Our study was designed to address this real-world scenario. By selecting time points of 4 and 8 hours, which are relevant within a single workday, we aimed not only to identify best practices for temporary preservation but also to investigate the specific reasons behind the poor RNA quality often resulting from improper handling during these delayed intervals.  

Line 38-40 In particularly, high-quality total RNA extracted from tissue is pivotal for advanced transcriptomic profiles using RNA molecular, such as RNA hybridization, real-time fluorescent quantitative polymerase chain reaction (qRT-PCR), bulk RNA microarrays, RNA-seq, and recently emerged spatial transcriptomics [1].

Comment: Regarding ‘recently emerged spatial transcriptomics’: The concept of spatial transcriptomics is old (+50 years) while combining in situ hybridization with Next-generation-Sequencing for a high-resolution view of the transcriptome tracks nine years back. The latter relies on e.g., machine learning to "deconvolute" bulk RNA-seq data for prediction of spatial gene expression. Maybe the authors could rephrase to ensure clarity for the readers?

Response 5:We are grateful to the reviewer for their valuable comments. which have helped us better understand spatial transcriptomics. We have revised the manuscript accordingly. (Line 42)

 Sometimes sentences seem out of context and appear to be without references, e.g.,:

  • Line 46-50: Only a few studies have compared the impact of both post-collection interval and different RNA preservative [8] on the RNA integrity in different tissues [13.]. In a previous study, we surveyed the effect of ex vivo ischemia time on RNA integrity and expression of genes isolated from 18 carcinoma tissues of 6 types of cancer. It is suggested that fresh tissues should be processed within 2 h to avoid RNA degradation. [14].
  • Line 171-172: Researchers found that RNA integrity affected the qRT-PCR performance, not the PCR efficiency. It recommends a RIN higher than 5 as good quality and higher than 8 as perfect for downstream application.

Comment: the authors need to rephrase the manuscript to ensure clarity and coherence for the reader and ensure that proper references are applied following scientific arguments stated in sentences by the authors.

Response 6:We sincerely thank the reviewer for this valuable feedback and have revised the manuscript as follows:

  • Updated the text (Lines 46-50): The sentence now reads: " Although previous studies have investigated the effects of RNA preservative types (Gambarino et al. 2024) (Camacho-Sanchez et al. 2013), ischemia time (Zheng et al. 2019), and tissue type (Walker et al. 2016) on RNA integrity, their simultaneous examination has been relatively scarce." (Lines 48-50)
  • Added a citation after Lines 171-172 as recommended. (Lines 192)

 The manuscript contains a high number of typos, and the grammar is very poor, e.g.:

  • Line 57-58: Given that most clinical tissues cannot be immediately pretreated and storage in liquid nitrogen in biobank lab upon collection due to the surgery complexity, shortage of liquid nitrogen and lack of timely delivery.
  • Line 63-64: Other two RNAlaters from Invitrogen and Sigma frequently adopted in biomedical studies were selected in this study.
  • Line 213: On the other hand, if the tissue sample has not been decided to conduct what kind of research at the collection in most clinical occasions.
  • Line 254-255: Consistent with previous reports, delayed preservation time has a negative correlation with RNA integrity for most tissue species.

Comment: it needs professional proof reading.

Response 7:Thank you for pointing out the need for professional proofreading. We agree that improving the language quality is crucial for the clarity and professionalism of the work. We have engaged the professional editing services to perform a comprehensive language polish of the entire manuscript. We believe the manuscript is now much improved and hope that the revised version meets with your approval.

  • Line 57-58: In clinical practice, the immediate cryopreservation of tissue samples in liquid nitrogen is often unfeasible owing to complex surgical procedures, a limited supply of liquid nitrogen, and logistical challenges in timely transportation. (Line 59-62)
  • Line 63-64: Two additional RNAlater reagents from Invitrogen and Sigma, which are commonly used in biomedical studies, were also selected for this investigation. (Line 66-67)
  • Line 213: In clinical practice, the specific downstream applications of a tissue sample are often not predetermined at the time of collection. In such cases, a multipurpose preservative such as RNAlater offers a distinct advantage, as it stabilizes RNA while also maintaining tissue integrity for histological analysis (Florell et al. 2001) (Suhovskih et al. 2019) and proteomic study (Bennike et al. 2016; Alyethodi et al. 2020) (Zhu et al. 2019).(Line 246-250)
  • Line 254-255: Consistent with previous reports, delayed preservation compromised RNA integrity in most tissue types examined. (Line 288-289)

In the discussion, the authors elaborate on the introduction with poor relevance to their own study: in line 179-191 the authors e.g. elaborate on spatial transcriptomics – an analysis the authors did not perform in their study. The authors could benefit from elaborating on the discussion of their own results. In general, the discussion is difficult to follow, and seems to lead to the promotion of the authors ‘upcoming quality management system’.

Comment: should this paper be accepted, the discussion needs to be revised with the purpose of promoting consistency between the relevance of the findings of their study on other studies discussed.

The authors do not directly refer to their results in the discussion. To improve the reading experience the authors should refer to relevant figure numbers in the discussion following sentences referring to their results.

Figure 3 should be revised, so that the colors indicating ‘Trizol’ and ‘Sigma’ appear more distinctive.

Response 8:We thank the reviewers for their valuable advice. We have thoroughly revised the discussion to enhance its consistency and to provide a more detailed elaboration of our own results. Additionally, in Figure 3, we have changed the colors representing 'Trizol' (to pink) and 'Sigma' to improve distinguishability.

Reviewer 2 Report

Comments and Suggestions for Authors

The manuscript "The Interference of RNA Preservative and Post-Collection Interval on RNA Integrity from Different Mice Tissues" by Xie et al. addresses a highly relevant and practical question in biospecimen science and biobanking. The study systematically investigates the combined effects of post-collection intervals and different RNA preservatives on the RNA Integrity Number (RIN) across ten mouse tissues. The topic is of significant importance for establishing standardized operating procedures (SOPs) in clinical and research biobanks. The experimental design is generally sound, and the findings provide valuable, tissue-specific guidance. However, the manuscript requires significant revisions, particularly in data presentation, statistical analysis clarity, and discussion, to fully support its conclusions and meet the standards for publication.

Major Revision Points

Data Presentation and Figure Clarity:

Figure 1: The linear regression plots are a good start, but they lack the underlying data points (e.g., individual RIN values for each biological replicate). It is impossible to assess the variance, the true strength of the correlations, or potential outliers. The figures must be revised to include scatter plots of the raw data.

Figure 3 (Crucial Figure): This is the core result of the paper, yet its current presentation is severely lacking.

Missing Statistics: The figure caption mentions significance markers (*, **, ***), but it is unclear what comparisons these refer to (e.g., vs. "None" at each time point? Between preservatives at each time point?). A two-way ANOVA (Preservative x Time) should be performed for each tissue, and the post-hoc results must be clearly indicated on the graphs.

Graphical Representation: The bar graphs, while common, may not be the best way to show the time-course data. Line graphs with points for each time and preservative might more effectively illustrate the trends. Most importantly, the absence of error bars (SEM, as mentioned in the methods) is a critical omission. The readers need to see the variability between the n=5 biological replicates.

Statistical Analysis Description and Application:

The Methods section (2.3) states that both one-way and two-way ANOVA were used, but it is not explicitly detailed which test was applied to which figure or dataset. For the key experiment (Fig. 3), a two-way ANOVA is the appropriate method to dissect the effects of the two main factors: Preservative and Time. The results of this analysis (e.g., main effects and interaction p-values) should be reported in the results text or figure legends.

The statement in the results (Page 6) that "Invitrogen RNAlater is more proper for the temporary storage of lung tissue and Sigma RNAlater is suit for intestine and uterus tissue... although no statistically significant difference was observed" is contradictory and problematic. If there is no statistical significance, this claim cannot be made. The authors must either provide statistical support or rephrase these conclusions to reflect the observed, but not statistically significant, trends.

Discussion:

The discussion would benefit from a deeper analysis of the biological reasons behind the observed tissue-specific differences. For instance:

Why are heart and ovary RNA so inherently stable? Is this related to lower baseline RNase activity, tissue structure, or metabolic rate?

Conversely, why is pancreas so susceptible to degradation, even in preservatives? The mention of pancreatic ribonucleases is correct but should be expanded and referenced earlier.

The discussion on the "overnight vs. immediate freezing" is valuable, but the hypothesis that it's due to small tissue size should be more strongly stated as a key limitation for extrapolation to larger human tissues.

The practical recommendation based on RIN thresholds (e.g., >5.5 for basic integrity, >7 for RNA-seq) is good, but the authors should explicitly state which of their experimental conditions met these thresholds for each tissue. A summary table would be immensely helpful for readers.

Minor Revision Points

Introduction:

The introduction is generally well-written. However, the transition from human biobanking challenges to the use of mouse tissues could be smoother. A sentence explicitly justifying the mouse model—not just due to difficulty obtaining human tissues, but also for the controlled conditions and genetic uniformity it offers for a methodological study like this—would be beneficial.

Materials and Methods:

Animal Model: The species is listed as "C57J". This is ambiguous. Please specify if it is C57BL/6J, the most common inbred strain. The sex of the mice is a critical variable, especially for hormone-responsive tissues like ovary and uterus. This must be stated.

Sample Size: "Each part of the tissue for RNA extraction used five mice as biological replicates." This is acceptable, but it should be clarified if the same five mice provided all ten tissues (a more powerful paired design) or if different mice were used for different tissues.

RNAlater Washing: The protocol for washing RNAlater-preserved tissues with PBS is mentioned. It would be useful to note if any pilot experiments were done to optimize the number and volume of washes to ensure complete removal without causing RNA loss.

Results:

Section 3.1: The description of "significant converse correlations" should be rephrased to "significant negative correlations."

Figure 2 Reference: In the text on Page 5, the results refer to "Figure 2A" and "Figure 2B," but the corresponding figure panels are not clearly labeled in the provided text. The final manuscript must ensure correct cross-referencing.

Clarity: The results text often describes what is in the figures without adding new interpretation. It can be streamlined to avoid redundancy (e.g., "As shown in Figure 3A, the RIN values of heart were the highest...").

Language and Formatting:

The manuscript requires thorough proofreading by a native English speaker or a professional service to correct grammatical errors and improve phrasing. Examples:

"not feasible due to delayed delivery" -> "challenging due to..."

"Owning to the difficulty" -> "Owing to the difficulty"

"It means that it is not necessary to wait for more time" -> "This indicates that it is unnecessary to wait longer".

The journal's citation format should be applied consistently.

Author Response

Point-by-point response to previous review comments

Dear Editors and Reviewers:

Thank you for your letter and for the Reviewers’ comments concerning our manuscript entitled “The Interference of RNA Preservative and Post-Collection Interval on RNA Integrity from Different Mice Tissues” (genes-3980191). Those comments are all valuable and very helpful for revising and improving our paper, as well as being of great importance and guiding significance to our research. We have carefully studied the comments and made corrections that we hope will meet with approval. The revised portion is marked in the revised manuscript. The spelling and grammar of the article have been professionally proofread and polished, with all revisions clearly highlighted in yellow. The main corrections and the point-by-point response to the reviewers’ comments are as follows:

Reviewer2

The manuscript "The Interference of RNA Preservative and Post-Collection Interval on RNA Integrity from Different Mice Tissues" by Xie et al. addresses a highly relevant and practical question in biospecimen science and biobanking. The study systematically investigates the combined effects of post-collection intervals and different RNA preservatives on the RNA Integrity Number (RIN) across ten mouse tissues. The topic is of significant importance for establishing standardized operating procedures (SOPs) in clinical and research biobanks. The experimental design is generally sound, and the findings provide valuable, tissue-specific guidance. However, the manuscript requires significant revisions, particularly in data presentation, statistical analysis clarity, and discussion, to fully support its conclusions and meet the standards for publication.

Major Revision Points

Data Presentation and Figure Clarity:

Figure 1: The linear regression plots are a good start, but they lack the underlying data points (e.g., individual RIN values for each biological replicate). It is impossible to assess the variance, the true strength of the correlations, or potential outliers. The figures must be revised to include scatter plots of the raw data.

Response 1:We thank the reviewer for pointing out this issue. As suggested, the individual RIN value for each biological replicate is displayed in Figure 3. This revised figure presents the data as a box plot combined with a scatter plot, which effectively illustrates the variance, potential outliers, and the distribution of all raw data points. The purpose of Figure 1 is to illustrate the trend of RNA integrity over time in different tissues without any preservative.

Figure 3 (Crucial Figure): This is the core result of the paper, yet its current presentation is severely lacking.

Missing Statistics: The figure caption mentions significance markers (*, **, ***), but it is unclear what comparisons these refer to (e.g., vs. "None" at each time point? Between preservatives at each time point?). A two-way ANOVA (Preservative x Time) should be performed for each tissue, and the post-hoc results must be clearly indicated on the graphs.

Response 2:We apologize for the lack of clarity in our original explanation. The significance markers (*, **, ***) in the bar graph itself represent the results of Tukey's multiple comparisons test, which compares each cell mean with every other cell mean within the same column, as generated by Prism. Additionally, the significance markers positioned above the chart (following the labels for time and preservative) indicate the overall statistical significance of the row factor (time) and the column factor (preservative).

Graphical Representation: The bar graphs, while common, may not be the best way to show the time-course data. Line graphs with points for each time and preservative might more effectively illustrate the trends. Most importantly, the absence of error bars (SEM, as mentioned in the methods) is a critical omission. The readers need to see the variability between the n=5 biological replicates.

Response 3:We thank the reviewer for this valuable suggestion. We agree that a line graph could effectively illustrate trends. During our initial analysis, we tested this approach but found that the scatter plots for each time point and preservative condition overlapped considerably, leading to a confusing and cluttered visual presentation. Therefore, we ultimately selected a bar graph format, which, while still capable of demonstrating the overall trend, more clearly displays the distribution of individual data points (scatter plots) and the associated error bars (SEM) for each group, as shown in the current Figure 2 and 3. Data represent means ± SEM,which has been supplemented in the figure legends.

Statistical Analysis Description and Application:

The Methods section (2.3) states that both one-way and two-way ANOVA were used, but it is not explicitly detailed which test was applied to which figure or dataset. For the key experiment (Fig. 3), a two-way ANOVA is the appropriate method to dissect the effects of the two main factors: Preservative and Time. The results of this analysis (e.g., main effects and interaction p-values) should be reported in the results text or figure legends.

 Response 4:We thank for your valuable suggestion. As requested, the detailed information regarding the statistical tests applied to the figures has been added to the Methods section(Line118-120). Furthermore, the results of these analyses, including p-values for main effects and interactions, have been expanded upon in the Results section and the corresponding figure legends. The interaction p-values have been added in the figure legends. (Line174-181)

The statement in the results (Page 6) that "Invitrogen RNAlater is more proper for the temporary storage of lung tissue and Sigma RNAlater is suit for intestine and uterus tissue... although no statistically significant difference was observed" is contradictory and problematic. If there is no statistical significance, this claim cannot be made. The authors must either provide statistical support or rephrase these conclusions to reflect the observed, but not statistically significant, trends.

Response 5:We sincerely thank for this valuable feedback and agree with the reviewer that the original statement was misleading and thank them for this insight. We have rephrased the claim accordingly and modified the results section to ensure it accurately reflects the observed trends. (Line170-171)

Discussion:

The discussion would benefit from a deeper analysis of the biological reasons behind the observed tissue-specific differences. For instance:

Why are heart and ovary RNA so inherently stable? Is this related to lower baseline RNase activity, tissue structure, or metabolic rate?

Conversely, why is pancreas so susceptible to degradation, even in preservatives? The mention of pancreatic ribonucleases is correct but should be expanded and referenced earlier.

The discussion on the "overnight vs. immediate freezing" is valuable, but the hypothesis that it's due to small tissue size should be more strongly stated as a key limitation for extrapolation to larger human tissues.

The practical recommendation based on RIN thresholds (e.g., >5.5 for basic integrity, >7 for RNA-seq) is good, but the authors should explicitly state which of their experimental conditions met these thresholds for each tissue. A summary table would be immensely helpful for readers.

Response 6:The discussion has been modified with a deeper analysis of the biological reasons behind the observed tissue-specific differences.

Why are heart and ovary RNA so inherently stable? Is this related to lower baseline RNase activity, tissue structure, or metabolic rate?--(Line204-212)

Conversely, why is pancreas so susceptible to degradation, even in preservatives? The mention of pancreatic ribonucleases is correct but should be expanded and referenced earlier.-- (Line213-219)

The discussion on the "overnight vs. immediate freezing" is valuable, but the hypothesis that it's due to small tissue size should be more strongly stated as a key limitation for extrapolation to larger human tissues. --(Line231-235)

The practical recommendation based on RIN thresholds (e.g., >5.5 for basic integrity, >7 for RNA-seq) is good, but the authors should explicitly state which of their experimental conditions met these thresholds for each tissue. A summary table would be immensely helpful for readers. --(We thank the reviewer for this valuable suggestion. However, given the complexity of our experimental design—which involves multiple tissue types, three time points, and three different preservatives—presenting specific RIN thresholds in a table would make it challenging to clearly convey the results for each condition. Therefore, to better distinguish RNA integrity levels, we have included reference lines at RIN 5.5 and 7 in all the figures and supplemented the result interpretation in result 3.3 (Line160-167)).

Minor Revision Points

Introduction:

The introduction is generally well-written. However, the transition from human biobanking challenges to the use of mouse tissues could be smoother. A sentence explicitly justifying the mouse model—not just due to difficulty obtaining human tissues, but also for the controlled conditions and genetic uniformity it offers for a methodological study like this—would be beneficial.

Response 7:We thank the reviewer for this insightful suggestion. The corresponding statement in the introduction has been modified to better reflect the perspective they highlighted. “We employed mouse tissues in this study due to the dual advantages of this model: the ability to maintain controlled conditions and genetic uniformity, which circumvent the challenges of obtaining high-quality fresh human tissues.” (Line73-75)

Materials and Methods:

Animal Model: The species is listed as "C57J". This is ambiguous. Please specify if it is C57BL/6J, the most common inbred strain. The sex of the mice is a critical variable, especially for hormone-responsive tissues like ovary and uterus. This must be stated.

Response 8:We appreciate the reviewer for pointing out the need for greater accuracy in our description of the animal model. We have now revised the text to provide a more precise account in the Methods section. (Line 88)

Sample Size: "Each part of the tissue for RNA extraction used five mice as biological replicates." This is acceptable, but it should be clarified if the same five mice provided all ten tissues (a more powerful paired design) or if different mice were used for different tissues.

Response 9:Following the reviewer's suggestion, we have ensured that all ten tissue samples were obtained from the same five mice. To this end, we pooled the tissues accordingly to minimize inter-individual variation. (Line 89-90)

RNAlater Washing: The protocol for washing RNAlater-preserved tissues with PBS is mentioned. It would be useful to note if any pilot experiments were done to optimize the number and volume of washes to ensure complete removal without causing RNA loss.

Response 10:Yes, pilot experiments were designed to optimize the number and volume of washes with PBS to ensure complete removal without causing RNA loss. Tissues immersed in Invitrogen and Sigma RNA preservative were washed with 1ml PBS per time and wash three times to remove the preservatives.

Results:

Section 3.1: The description of "significant converse correlations" should be rephrased to "significant negative correlations."

 Response 11:We have revised the text accordingly to address this point. (Line 126)

Figure 2 Reference: In the text on Page 5, the results refer to "Figure 2A" and "Figure 2B," but the corresponding figure panels are not clearly labeled in the provided text. The final manuscript must ensure correct cross-referencing.

Response 12:We apologize for the error in labeling within the Results section. This has now been corrected. (Line 157-160)

Clarity: The results text often describes what is in the figures without adding new interpretation. It can be streamlined to avoid redundancy (e.g., "As shown in Figure 3A, the RIN values of heart were the highest...").

Response 13:In response to the suggestion, we have thoroughly revised the Results section to streamline descriptive statements and, more importantly, to incorporate meaningful interpretation and discussion of the findings. (Line 157-166)

Language and Formatting:

The manuscript requires thorough proofreading by a native English speaker or a professional service to correct grammatical errors and improve phrasing. Examples:"not feasible due to delayed delivery" -> "challenging due to..."

"Owning to the difficulty" -> "Owing to the difficulty"

"It means that it is not necessary to wait for more time" -> "This indicates that it is unnecessary to wait longer".

The journal's citation format should be applied consistently.

Response 14:We are grateful for the reviewer's professional suggestions. In response, we have proofread the manuscript by a professional service to correct grammatical mistakes, polished the language for better clarity, and carefully standardized the citations to ensure full compliance with the journal's format.

"not feasible due to delayed delivery" -> "challenging due to..." (Line 15)

"Owning to the difficulty" -> "Owing to the difficulty"(modified)

"It means that it is not necessary to wait for more time" -> "This indicates that it is unnecessary to wait longer”. (Line146-148)

Reviewer 3 Report

Comments and Suggestions for Authors

The manuscript entitled "The Interference of RNA Preservative and Post-Collection Interval on RNA Integrity from Different Mice Tissues" investigates different RNA preservatives under room temperature conditions for up to 8 hours. I consider it a very important research in the field, and its results are relevant.

However, in my opinion, there is a lack of existing information in the bibliography to include in the introduction as background and in the discussion to compare with the results of the current work. These are some examples:

Camacho-Sanchez M, et al. 2013, https://doi.org/10.1111/1755-0998.12108 PMID: 23617785

Florell SR, et al. 2001, https://doi.org/10.1038/modpathol.3880267

Bennike TB, et al.2016, https://doi.org/10.1016/j.euprot.2015.10.001 

Latorre N, et al. 2024, https://doi.org/10.1371/journal.pone.0314013

Author Response

Point-by-point response to previous review comments

Dear Editors and Reviewers:

Thank you for your letter and for the Reviewers’ comments concerning our manuscript entitled “The Interference of RNA Preservative and Post-Collection Interval on RNA Integrity from Different Mice Tissues” (genes-3980191). Those comments are all valuable and very helpful for revising and improving our paper, as well as being of great importance and guiding significance to our research. We have carefully studied the comments and made corrections that we hope will meet with approval. The revised portion is marked in the revised manuscript. The spelling and grammar of the article have been professionally proofread and polished, with all revisions clearly highlighted in yellow. The main corrections and the point-by-point response to the reviewers’ comments are as follows:

Reviewer 3

The manuscript entitled "The Interference of RNA Preservative and Post-Collection Interval on RNA Integrity from Different Mice Tissues" investigates different RNA preservatives under room temperature conditions for up to 8 hours. I consider it a very important research in the field, and its results are relevant.

However, in my opinion, there is a lack of existing information in the bibliography to include in the introduction as background and in the discussion to compare with the results of the current work. These are some examples:

Camacho-Sanchez M, et al. 2013, https://doi.org/10.1111/1755-0998.12108 PMID: 23617785

Florell SR, et al. 2001, https://doi.org/10.1038/modpathol.3880267

Bennike TB, et al.2016, https://doi.org/10.1016/j.euprot.2015.10.001 

Latorre N, et al. 2024, https://doi.org/10.1371/journal.pone.0314013

Response:We thank the reviewer for this valuable comment. We have carefully reviewed the relevant literature and have revised the introduction and discussion section to incorporate a comparison of these findings with the results of our current work. The corresponding references have been added to the bibliography.

Camacho-Sanchez M, et al. 2013, https://doi.org/10.1111/1755-0998.12108 PMID: 23617785 (Line 49 [8])

Preservation of RNA and DNA from mammal samples under field conditions

Florell SR, et al. 2001, https://doi.org/10.1038/modpathol.3880267

Preservation of RNA for Functional Genomic Studies: A Multidisciplinary Tumor Bank Protocol(Line 250 [32] )

Bennike TB, et al.2016, https://doi.org/10.1016/j.euprot.2015.10.001 

Comparing the proteome of snap frozen, RNAlater preserved, and formalin-fixed paraffin-embedded human tissue samples(Line 250 [34])

Latorre N, et al. 2024, https://doi.org/10.1371/journal.pone.0314013

RNA quality and protamine gene expression after storage of mouse testes under different conditions (Line 212 [29])

Round 2

Reviewer 2 Report

Comments and Suggestions for Authors

You have answered all my objections. It's fine with me.

Reviewer 3 Report

Comments and Suggestions for Authors

I consider the manuscript to have improved substantially, and I recommend its publication in its current form.